# A Classification Model of EEG Signals Based on RNN-LSTM for Diagnosing Focal and Generalized Epilepsy

**DOI:** 10.3390/s22197269

**Published:** 2022-09-25

**Authors:** Tahereh Najafi, Rosmina Jaafar, Rabani Remli, Wan Asyraf Wan Zaidi

**Affiliations:** 1Department of Electrical, Electronics and Systems Engineering, Universiti Kebangsaan Malaysia, Bangi 43600, Malaysia; 2Department of Medicine, Hospital Canselor Tuanku Muhriz, Universiti Kebangsaan Malaysia, Cheras, Kuala Lumpur 56000, Malaysia

**Keywords:** electroencephalography (EEG), epilepsy, long short-term memory (LSTM), theta frequency band, longitudinal bipolar montage (LB), signal processing, classification

## Abstract

Epilepsy is a chronic neurological disorder caused by abnormal neuronal activity that is diagnosed visually by analyzing electroencephalography (EEG) signals. Background: Surgical operations are the only option for epilepsy treatment when patients are refractory to treatment, which highlights the role of classifying focal and generalized epilepsy syndrome. Therefore, developing a model to be used for diagnosing focal and generalized epilepsy automatically is important. Methods: A classification model based on longitudinal bipolar montage (LB), discrete wavelet transform (DWT), feature extraction techniques, and statistical analysis in feature selection for RNN combined with long short-term memory (LSTM) is proposed in this work for identifying epilepsy. Initially, normal and epileptic LB channels were decomposed into three levels, and 15 various features were extracted. The selected features were extracted from each segment of the signals and fed into LSTM for the classification approach. Results: The proposed algorithm achieved a 96.1% accuracy, a 96.8% sensitivity, and a 97.4% specificity in distinguishing normal subjects from subjects with epilepsy. This optimal model was used to analyze the channels of subjects with focal and generalized epilepsy for diagnosing purposes, relying on statistical parameters. Conclusions: The proposed approach is promising, as it can be used to detect epilepsy with satisfactory classification performance and diagnose focal and generalized epilepsy.

## 1. Introduction

Epilepsy is a chronic disorder inducing subjects to experience seizures, leading to cognitive impairments, medical and psychiatric comorbidities, social stigmatization, and, in general, poor quality-of-life (QOL) [1]. A recent study reported that the prevalence of lifetime epilepsy was 7.8 per 1000 individuals in Malaysia in 2021 [2]. Diagnoses of epilepsy are basically clarified by an epileptologist based on a clinical assessment, neuro imaging, and the visual detection of interictal epileptiform discharges (IEDs) appearing in 30% of cases in their electroencephalography (EEG) signals [3]. EEG reveals a general overview of neuronal activity in disparate cortical regions by representing potential differences between certain areas of the brain and a determined reference on the head surface in timeseries data [4]. According to [5], known epilepsy is classified into two categories based on the clinical symptoms and the localization of manifested abnormalities in EEG. These epilepsy categories are focal epilepsy, which involves the partial region of the brain, and generalized epilepsy, which affects all regions of the brain. Although anti-seizure drugs (ASDs) are vastly used to control the number of seizures, about one-third of epileptic patients in the world are refractory to treatment, and surgical operation in which the epileptogenic foci need to be removed is the only option. As a result, detecting the affected area linked with the seizure onset zone plays a pivotal role in the process of treatment [6]. The challenge to the diagnosis phase mostly arises from the need to assess long-term EEG recordings, which is time-consuming and prone to inaccuracy due to human error. Consequently, training models for IED observation may be useful in the diagnosis process, especially at times when an epileptologist is unavailable.

Literature shows that the focus in the majority of epilepsy studies is summarized in seizure detection using machine or deep learning techniques to determine the type of epilepsy [7]. This study is concentrated on interictal duration and in the cases in which IEDs are not necessarily available. Diagnosing different types of epilepsy does not solely depend on analyzing EEG signals for discovering IEDs. In this regard, machine learning techniques are used in analyzing epileptic EEG signals. The analysis comprises the following main steps: pre-processing, feature extraction and classification.

Feature extraction of machine learning technique is done in time, frequency, or time–frequency domains [8]. Time–frequency methods such as flexible analytic wavelet transform [9,10], short-time Fourier transform [11], discrete wavelet transform (DWT) [12], Hilber Huang transform [13], and empirical mode decomposition [14] have been considered for diagnosing epilepsy. Automatic focal and non-focal epilepsy were detected using entropy-based features from flexible analytic wavelet transform in [10]. Wavelets, scatter matrices, and quadratic classifiers were, respectively, employed for feature extraction, feature dimensionally reduction, and classification in [15] in order to classify EEG signals to detect epileptic seizures. The study reached a 99% accuracy in distinguishing healthy controls from subjects with epilepsy, with or without seizures. An interictal seizure-free period has been analyzed by [16] using triggering signals of intermittent photic stimulation (IPS) reporting frequency domain features; the theta band is the most fitting feature in diagnosing generalized epilepsy in the visual cortex. This classification has been done with a support vector machine (SVM) in 18 Hz IPS, reaching the best discrimination between groups. The authors in [17] in 2017 introduced a statistical-based solution to overwhelm the empirical or arbitrarily determination of the level of decomposition in wavelets. The study reached an accuracy of more than 80% in detection using an SVM for two datasets: Bern Barcelona and University of Bonn. The authors in [18] evaluated different wavelet families using a probabilistic neural network (PNN) and an SVM. The study reported Coiflet as the best wavelet family in diagnosing epilepsy. Literature shows that the SVM is a valuable tool and is vastly used as a valuable classifier in a variety of clinical diagnostic research [19].

After feature extraction, feature dimensionality reduction is a vital step in analyzing signals in cases where we want to reduce irrelevant features and determine the most effective ones with a high model performance. This can be done through various methods such as feature selection or a combination of features, both of which rely on mathematical solutions behaving as filters, wrappers, and embedded strategies [20,21]. Detecting epileptic seizures with a focus on feature selection based on fuzzy membership was achieved in [22]. The authors in [23] conducted a comparative study to analyze discriminative features using various feature selection techniques in epilepsy. A method for EEG feature selection was introduced in [24] via stacked deep embedded regression with joint sparsity.

Classification steps have been taken by a variety of linear and non-linear classifiers, such as the decision tree [25], logistic regression [26], the k-nearest neighbor, the support vector machine [27], Naive Bayes [28], and artificial neural networks [29,30,31], or deep learning [32]. Artificial intelligence encompasses a variety of areas, and one of them is deep learning (DL). Before the rise of DL, conventional machine learning algorithms involving feature extraction were used. Their performance was limited to the ability of those handcrafting the features. However, in DL, the extraction of features and classification are entirely automated. These techniques have made significant advances in many areas of medicine, such as in the diagnosis of epileptic seizures.

The drawback of machine learning algorithms, albeit still beneficial, is that their performance is limited to the ability of those handcrafting the features [33]. Artificial intelligence encompasses a variety of areas, and one of its branches is deep learning. The priority of deep learning compared to machine learning is that the extraction of features and classification are entirely automated. The paper cited above comprehensively revies deep learning techniques in epilepsy studies from 2016 to 2021 using EEG and neuroimaging techniques with a focus on seizure detection. The review represented the available epilepsy datasets, such as Freiburg, CHB-MIT, and Bonn. It was reported that the majority of researchers employed their own clinical dataset. With a focus on EEG and epilepsy, the paper represented the definition of some of the high-usage deep learning models: one-dimensional convolutional neural networks (1D-CNNs), recurrent neural networks (RNNs) and two of its branches, long short-time memory (LSTM) and gated recurrent units (GRUs), autoencoders (AEs), CNN-RNNs, and CNN-AEs. The paper collected 24 papers on EEG signals in epilepsy detection using 1D-CNNs. The studies used 4–33 layers for their models and found diagnosis accuracies ranging from 79.34% to 99.28% (with one reaching 100%) using mostly Softmax and in some cases SVM classifiers. In 15 studies that applied an RNN and its different branches, mainly LSTM, the accuracy reported was superior (from 84.35% to 98.91% and one 100%) to the CNN. The papers mostly used 4 layers (minimum 3, maximum 48 layers) for their models and mainly used using Softmax and Sigmoid, with one study using multilayer perceptron (MLP) classifiers.

An RNN is developed to process timeseries data through cyclic connections based on feedforward neural networks. The method has been vastly used for seizure prediction with classification approaches. The history of input in an RNN is mapped in order to predict each output by weighting the temporal relationships between the data at each time point. The issue is that a vanishing gradient problem causing the given input influences hidden and output layers, thus decaying or exploding exponentially over time [34]. One of the popular solutions for this is to use LSTM. RNN-LSTM consists of connected subnetworks called a memory block, which remembers inputs for a long time. The authors in [35] used a combination of a 1D-CNN and LSTM for epileptic seizure detection. The authors in [36] investigated the automatic detection of epilepsy by a CNN-LSTM using the University of Bonn dataset and reached an accuracy of more than 80%. CNN-LSTM was further used in [37] on the same dataset to detect epileptic seizures, approaching a 99.71% accuracy, with a focus on the Tunable-Q Wavelet Transform (TQWT) in feature extraction. Using time series data and LSTM to analyze sequenced data, the authors in [38] were able to introduce a hybrid model by a dense convolutional network and LSTM using information transferred from DWT to images for prediction purposes.

In the present study, a model using RNN-LSTM is proposed for distinguishing normal subjects from subjects with epilepsy without observing IEDs. The model is further validated by correctly diagnosing focal and generalized epilepsy. Hence, retrospective EEG data from normal subjects and patients with focal and generalized epilepsy were used. The data of normal subjects and focal epilepsy patients were used for classification purposes, whereas the data of focal and generalized epilepsy patients were used for validating our classification model. After pre-processing using DWT, a longitudinal bipolar (LB) montage was calculated for all groups. Next, features were extracted in time and frequency domains for further selection based on *p*-values of Pearson’s linear correlation coefficient. Signals were segmented, and the network was trained by a sequence of selected features extracted from each segment. Instead of raw signals, we used features extracted from segments as sequenced data to feed the network. The optimal group of features and the best model are employed to diagnose focal and generalized groups via classifying their LB channels as epileptic or normal, as depicted in Figure 1. The findings reveal that the proposed classification model is effective in detecting epileptic signals from normal signals and diagnosing focal and generalized epilepsy.

## 2. Materials and Methods

### 2.1. Dataset

In this study, two sets of data were used for two purposes: to train the classification model (classification approach) and to validate the model (diagnosing approach). EEG data were collected from the hospital Canselor Tuanku Muhriz (HCTM) in Cheras, Malaysia. In the classification approach, the focus is classification between normal and epileptic subjects. In this regard, the temporal lobe channels of 42 patients suffering from non-lesional temporal lobe epilepsy (TLE) and the temporal lobe channels of 62 normal subjects were used. In the diagnosing approach, i.e., distinguishing between focal and generalized epilepsy, whole EEG channels of 50 patients with generalized epilepsy and whole EEG channels of 42 TLE patients were used. Some generalized patients had a normal EEG and were diagnosed as generalized epilepsy patients based on clinical symptoms. EEG was recorded from Fp1, Fp2, F3, F4, F7, F8, C3, C4, P3, P4, T3, T4, T5, T6, O1, and O2: a total of 16 electrodes via the Nicolet EEG device based on the 10–20 EEG standard electrode placement system for each case. For the measurement of EEG signals, subjects (male and female; age: 36.90 ± 13.40) were prepared to contain a contact impedance of less than 5 kΩ and recorded at a 500 Hz sample rate, and data recording was done during a resting state. The affected channels in the TLE cases were deemed as the epilepsy group, and the same channels in the normal cases were deemed as the normal group. From EEG reports and patients’ clinical profiles, we determined that the placements of the affected channels were identified by neurologists in the inferior, mid, and superior areas of the temporal gyrus, i.e., in the right, left, or both hemispheres reflected in specific EEG channels under LB montage: Fp1-F7, F7-T3, T3-T5, T5-O1, Fp2-F8, F8-T4, T4-T6, and T6-O2. The affected channels were referred to as the channels with epilepsy localization. Therefore, the montage was calculated for 10 s of all datasets, 51 affected channels were identified as the epilepsy group, and 62 channels of the same regions were considered as the normal group. This data were studied to train our network. In the final stage, the classification model was validated by testing all LB channels from 50 generalized patients and the same 42 TLE patients (lateralized TLE and both hemispheres affected). The classification model and all EEG analysis were implemented via MATLAB (R2020a).

### 2.2. Pre-Processing

Pre-processing focused on signal preparation in the aspect of eliminating artifacts due to muscular movement and blinking as well as swallowing manually. DWT using coif3 from the Coiflet family was applied to 10 s of raw signals for both the epileptic and the normal groups to eliminate power line noise by three levels of signal decomposition [39]. A longitudinal bipolar montage was calculated for each group by subtracting the amounts of potential differences between pertinent electrodes [40]. Figure 2 demonstrates the LB montage and the calculation details. In the figure, the cross mark represents the placement of the reference (Ref) while recording EEG signals—somewhere between the frontal lobe and central sulcus. LB consists of 18 channels; in this study, only 16 channels in the left and right posterior and anterior regions were calculated, and the leads Fz, Cz, and Pz were ignored.

LB calculation is defined by replacing targeted leads, as shown by arrows in Figure 2, with the original reference during the EEG recording. For instance, for calculating Fp1-F7 as the first LB channel, first the potential difference recorded from both leads must be added by the amount of the original Ref. The new F7 should then be considered as the new reference for Fp1. This means that the value of F7 must be subtracted from the value of Fp1. In this study, due to the deficiency of the original Ref value, this amount was considered as a common subtracted value in LB calculation. Figure 3 represents 10 s of normal and epileptic EEG signals from one LB channel in the temporal region. Figure 4 exhibits a sample of de-noised generalized and TLE EEG data based on an LB montage.

### 2.3. Feature Extraction

Fifteen features in the time and frequency domains were extracted from each channel of both the epileptic and normal group: mean, standard deviation (STD), peak-to-peak (P2P), min, max, skewness (Skew), kurtosis (Kurt), peak-to-root sum square (P2RMS), root sum square (RSS), power of delta frequency band (delta; 1–4 Hz), power of theta frequency band (theta; 4–8 Hz), power of alpha frequency band (alpha; 8–14 Hz), power of beta frequency band (beta; 14–30 Hz), and power of gamma frequency band (gamma; over 30 Hz). Figure 5 exhibits a sample of the power spectrum density for one epileptic channel and one normal channel.

### 2.4. Feature Selection

Pearson’s rank correlation coefficients between all pairs of variables were calculated. The hypothesis test was considered in order to determine which correlations are significantly different from zero. Features with *p*-value <0.05, i.e., theta, alpha, beta, mean, min, skew, and kurt, were considered if they show a high classification performance. In addition, the features with the lowest correlation (≤20%) were considered, and the features with a high correlation (≥80%) were added separately to the group. Therefore, three groups of five features with low correlations were considered to feed the network.

### 2.5. Classification Model Using RNN-LSTM

In this research, we used the RNN-LSTM architecture to identify epilepsy and diagnose focal and generalized epilepsy. Hence, the network was implemented with five layers: a sequence input layer, a bidirectional LSTM (BiLSTM) layer with 200 hidden units, a fully connected layer, a SoftMax layer, and a classification output layer. Table 1 represents the details of deep learning layers, values, and descriptions for training the network. The network is fed based on 10-fold cross validation achieved by 80% of the data for training and the remaining 20% was used for testing. EEG signals were segmented with 50% overlapping—1 s for each. Three groups of features were extracted from each segment. The model was trained by each group of features. The model with the best performance was considered as our classification model. In the next stage of the study, the classification model was applied to all LB channels of the focal and generalized epilepsy groups. The overall value of infections for each channel for each group was then calculated, separately. The variance of the overall values, which shows that the channel is affected, was measured for each channel of both groups. A high variance indicates that some channels are affected more than others by our model. Focal and generalized epilepsy cases were encountered when features had high and low variance; respectively. This was used as to validate our classification model.

## 3. Results and Discussion

### 3.1. Classification Approach

Figure 6a shows the significance level for the correlation tests specified as a scalar between 0 and 1, representing a low or high correlation, respectively. Negative values show a negative correlation among the relevant features. The figure shows the correlation between two features in the group. As shown, there are three strong correlated group of features surrounded by red boxes that need to be chosen individually, while the rest in these boxes are dropped out [41]. In addition, there are seven features where *p* < 0.05, i.e., theta, alpha, beta, mean, min, skew, and kurt, indicating significant features for discrimination (Figure 6b). In addition, a high correlation between the power of frequency bands restricted us from introducing three groups of features separated by theta, alpha, or beta. Therefore, we will have three groups of features with theta, alpha, and beta added separately to the mean, min, skew, and kurt.

The ability of the model performance for discrimination has been characterized by sensitivity, specificity, accuracy, positive predictive value (PPV), and negative predictive value (NPV), referring to Equations (1)–(5), respectively. The sensitivity presents the percentage of detecting case subjects, while specificity emphasizes the ability to detect normal subjects. The accuracy is the amount of total detection for both patients and normal subjects from the study population. PPV and NPV represent the proportion of subjects with a positive test result who actually have the disease and those with a negative result who do not have the disease, respectively.
(1)$$\mathrm{Sensitivty}=\frac{\mathrm{TP}}{\mathrm{TP}+\mathrm{FN}}\times 100$$
(2)$$\mathrm{Specificity}=\frac{\mathrm{TN}}{\mathrm{TN}+\mathrm{FP}}\times 100$$
(3)$$\mathrm{Accuracy}=\frac{\mathrm{TP}+\mathrm{TN}}{\mathrm{TN}+\mathrm{FP}+\mathrm{TP}+\mathrm{FN}}\times 100$$
(4)$$\mathrm{PPV}=\frac{\mathrm{TP}}{\mathrm{TP}+\mathrm{FP}}\times 100$$
(5)$$\mathrm{NPV}=\frac{\mathrm{TN}}{\mathrm{FN}+\mathrm{TN}}\times 100$$
where TP is true positive, representing patients that were correctly diagnosed among the patients, TN is true negative, indicating the number of normal subjects that were correctly diagnosed among normal subjects, FN is false negative, showing the number of subjects wrongly diagnosed as normal among the patients, and FP is false positive, representing the number of subjects wrongly diagnosed as patients among the normal subjects. Table 2 presents the network performance for three groups of features. It seems that the discrimination ability using the first group, highlighted in the table, is higher than the rest. Table 3 shows the details of the confusion matrix for three groups of selected features. The table shows that the model can detect epilepsy well, but there is still some confusion in detecting some samples. The model based on the features of Group 1 incorrectly detected three samples as epileptic instead of normal, and it detected only one normal sample as epileptic. Similarly, with the features of Group 3, there were significantly more false negatives than there were false positives. In contrast, the features of Group 2 led to the opposite balance, where the number of normal samples detected as epileptic was slightly higher. In general, the model behaved better using the features of Group 1, with emphasis on the power of the theta frequency band.

### 3.2. The Diagnosing Approach

The classification model was applied to each LB channel of EEG recordings from subjects with generalized and focal epilepsy. Figure 7 illustrates the result of the classification model for both groups by presenting the percentage of affected channels in the left or right posterior or anterior LB montage. Regarding focal epilepsy, the majority of the left and right posterior LB channels were diagnosed as affected in over 60% of the study population. Moreover, in the same group, all anterior LB channels were classified as affected in approximately 50% of the study population. In contrast, the generalized epilepsy findings indicate that affected LB channels were affected at a slightly constant rate in approximately 50% of its population. The average (55.59%) and variance (272.14) of detection for focal epilepsy, and the average (44.01%) and variance (51.05) of detection for generalized epilepsy, validate the classification model as a promising tool for distinguishing epileptic signals from normal signals without analyzing IEDs.

Figure 7 appears to indicate that the temporal lobe is not the only region affected in TLE, and the frontal area may be involved in this type of epilepsy. TLE is associated with long-term memory dysfunction [42]. The frontal lobe is related to cognition comprising executive skills as well as memory [43]. Evidence from neurophysiological and neuroimaging literature confirms the deficiency in executive function in the frontal lobe and working memory in TLE cases. The facts support this part of our study achievement.

Turning back to feature extraction step, the power of the theta frequency band were more effective in the classification model compared to the alpha and beta frequencies, albeit being accompanied with four other features (mean, min, skew, and kurt). There is an assumption that epilepsy characteristics are related to theta band connectivity in patients suffering from epileptic seizures [44]. A systematic review confirmed a consistent association between the theta frequency band and idiopathic epilepsy [45]. The authors in [46] reported an increment in theta activity during resting states in patients with major epilepsy syndromes. A diagnostic epilepsy study worked on the spectral power of different frequency bands in controls, patients with well controlled idiopathic generalized epilepsy, and drug-resistant patients. The study confirmed a higher interictal EEG spectral power in all frequency bands, and the reported frequency band were useful in diagnosing epilepsy.

Furthermore, the hippocampus has been claimed as the main structure involved in generating theta oscillations [47]. In 2021, the authors in [48] assessed the effects of hippocampal stimulation by inducing theta frequency, resulting in convulsion elimination. The authors in [49] conducted a successful animal study using deep brain stimulation (DBS) in attenuating seizures in TLE. The hypothesis was the reversal of the effects of stimulation augmentation of the hippocampal theta oscillation. Moreover, it was reported that epileptic seizures occur less often during waking periods or paradoxical sleep period and in conditions when the hippocampal theta rhythm can be observed.

## 4. Conclusions

In this work, we defined two approaches in proposing a classification model for detecting epilepsy. The first approach was an essential one (the classification approach), and the second approach (the diagnosing approach) was focused on using the model to diagnose focal and generalized epilepsy using statistics. In the classification approach, brain signal processing techniques were implemented for signal pre-processing. De-noising signals and discovering affected channels were performed via DWT and LB montage calculation, respectively. Feature extraction was performed using methods in the time and frequency domains. Feature selection was performed using correlation coefficients. Classification was performed using RNN-LSTM. In this approach, the first aim is to find optimal features in distinguishing epileptic subjects from normal subjects, whereas the second aim is to extract features from segmented EEG signals. Continuous features are fed to the network. In the diagnosing approach, the best classification model was used for each LB channel of focal and generalized epilepsy subjects. The variance of the overall affected channels represented the type of epilepsy, where a high and low variance refers to focal and generalized epilepsy, respectively. In this work, three groups of EEG data were considered: normal subjects (non-epilepsy) and subjects with focal (TLE) and generalized epilepsy. Affected channels were collected from subjects without epilepsy and with focal epilepsy using the classification approach, whereas subjects with focal and generalized epilepsy were considered in the diagnosing approach. The results show that the best classification model was achieved through employing mean, min, skewness, kurtosis, and the power of the theta frequency band, with 96.70% accuracy, 94.44% sensitivity, and 97.6% specificity. Furthermore, it seems that the theta frequency band was more successful than alpha and beta in the detection procedure. The results show a remarkable difference in variance in the diagnosing approach via the proposed classification model. The most important limitation here is the potential lack of IEDs in epileptic EEG signals during interictal periods. Therefore, confident affected signals according to EEG reports were considered as a reference for training network. Furthermore, we could not test the model for other different types of focal lobe epilepsy due to the lack of data caused by a lower prevalence. In our study, we used affected signals from TLE cases because TLE is significantly prevalent in the HCTM dataset. TLE is also known as one of the most common causes of focal epilepsy and one of the most frequent indications of epilepsy surgery. Therefore, TLE became the only option for checking the validity of the model in the diagnosing approach. There is a thread to validate in the classification stage. Despite the selected features that significantly distinguished epileptic subjects from normal subjects, the validation may become confronted with a lower variance in diagnosing focal versus generalized epilepsy by increasing the amount of data. In future work, investigation on more data is suggested. Moreover, it would be beneficial not to combine both hemispherical focal epilepsy in the same diagnosing process. The model then might be needed to optimize the internal parameters that indicate the affected hemisphere.

## Figures and Tables

**Figure 1 sensors-22-07269-f001:**
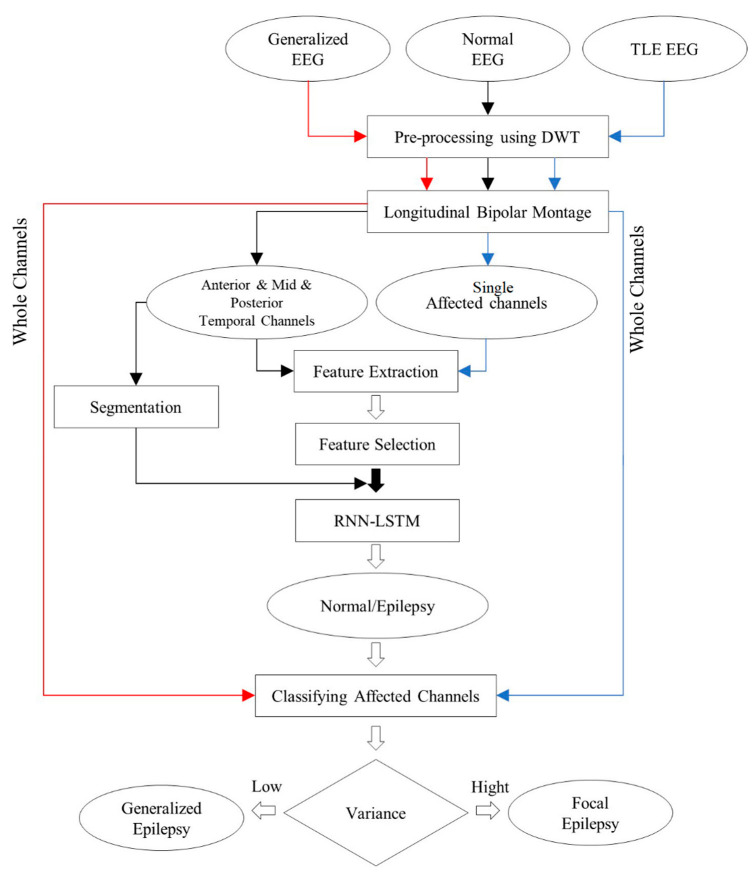
A flowchart of the study.

**Figure 2 sensors-22-07269-f002:**
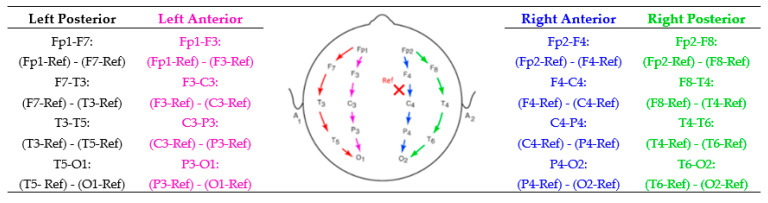
Longitudinal bipolar montage calculation separated in the left and right posterior and anterior areas.

**Figure 3 sensors-22-07269-f003:**
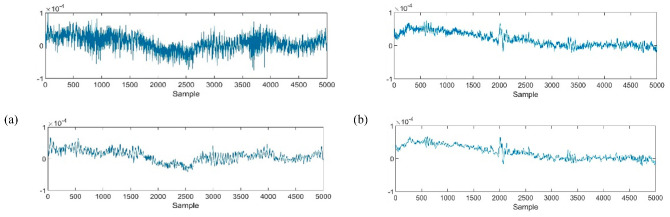
Samples of raw signals (top) and de-noised signals (down) of normal (**a**) and epileptic (**b**) signals recorded from T4−T6. The X-axis shows the potential difference (µv).

**Figure 4 sensors-22-07269-f004:**
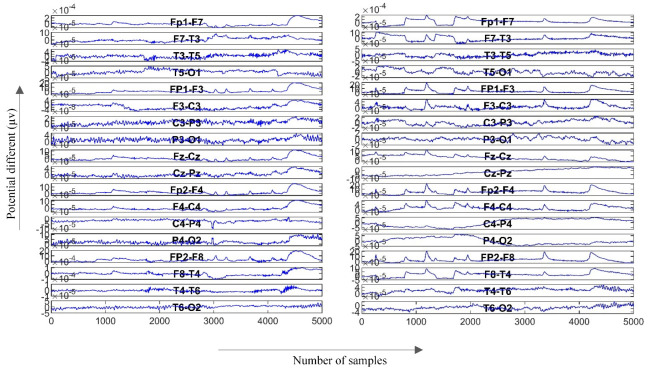
Generalized epilepsy (**left**) and TLE (**right**) samples based on the LB montage.

**Figure 5 sensors-22-07269-f005:**
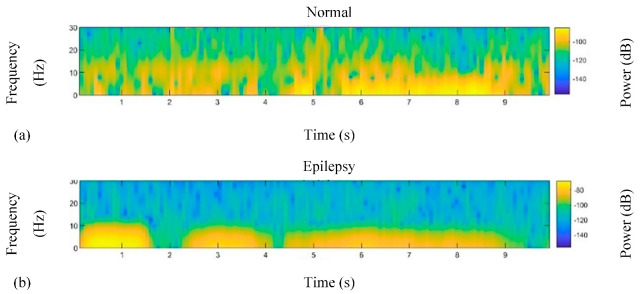
A Sample of power spectral density for one normal channel (**a**) and one epileptic (**b**) channel.

**Figure 6 sensors-22-07269-f006:**
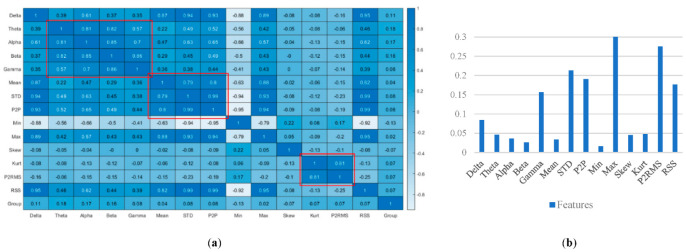
Correlation coefficient among features (**a**), *p*-value for each feature in group (**b**).

**Figure 7 sensors-22-07269-f007:**
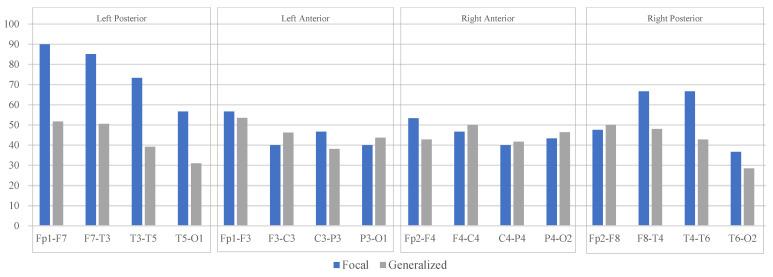
The result of the classification model for focal (blue) and generalized (grey) groups in classifying each channel as affected channels. The X-axis represents LB channels categorized in the left and right posterior and anterior areas. The Y-axis represents the percentage of affected channels according to the population of each group.

**Table 1 sensors-22-07269-t001:** Deep learning layers and network training options.

Deep learning Layers		Value	Description
BiLSTMLayer		BiLSTM with 200 hidden units	Output Mode: Last
FullyConnectedLayer		2 fully connected layers	
SoftmaxLayer		Softmax	
ClassificationLayer		Crossentropyex	
**Training Option**	**Value**	**Description**	
ADAM	-	Adoptive moment estimation—Optimization Algorithm	
MaxEpochs	30	30 passes through the training data in the network	
MiniBatchSize	150	Leads the network to look at 150 training signals at a time	
InitialLearnRate	0.01	Assists to speed up the training process	
GradientThreshold	1	To stabilize the training process by preventing gradients from becoming too large	

**Table 2 sensors-22-07269-t002:** The performance of the network in distinguishing normal subjects from epileptic subjects in the training stage.

Groups of Features	Acc (%)	Sen (%)	Spc (%)	PPV (%)	NPV (%)
Group 1: mean, min, skew, kurt, theta	96.1	96.8	97.4	98.4	92.7
Group 2: mean, min, skew, kurt, alpha	90.4	94.0	85.4	91.0	89.7
Group 3: mean, min, skew, kurt, beta	91.4	87.3	97.6	98.0	82.0

**Table 3 sensors-22-07269-t003:** The details of the confusion matrix for the three groups of features.

		Predicted Classes					
		Group 1		Group 2		Group 3	
		Epileptic	Normal	Epileptic	Normal	Epileptic	Normal
**Actual Classes**	**Epilepsy**	38	1	35	6	40	1
	**Normal**	3	62	4	59	8	55

## Data Availability

The data used in the study are not publicly available as the data repository belongs to HCTM that bounds to the ethics approval.

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
