# Peer review of "A Classification Model of EEG Signals Based on RNN-LSTM for Diagnosing Focal and Generalized Epilepsy"

_sensors, 2022, doi:10.3390/s22197269_

Round 1
Reviewer 1 Report
1. A motivation why a model using RNN-LSTM deep learning network is required. Present a discussion of pros and cons. Also, discuss other possible alternatives.
2. The overview of related works in the introduction section lacks of structure and organization. I suggest to discuss the review papers which summarized multiple other studies on the topic of this paper such as, for example, “A review on the pattern detection methods for epilepsy seizure detection from EEG signals”, “Epilepsy detection from EEG signals: a review”, and “Automated epileptic seizure detection in pediatric subjects of chb-mit eeg database—a survey”. Then you can proceed with the discussion on individual studies, and finally summarize the limitations of other studies as a motivation of your research.
3. Did you apply any filtering of the EEG signals to remove muscular movement and eye blinking artifacts?
4. How do you do with colinear features?
5. Present the structure of the proposed deep learning model as a table with all parameters of each layer presented.
6. Explain your cross-validation procedure in more detail. If part of the measurements of a subject are included in the training set and if the other measurements of that same subject are in the testing set, this will overestimate the performance!
7. The statistical analysis of the results must be included. For example, are there any statistically significant difference between the results achieved using theta, alpha and betta bands? Perform statistical testing to confirm or reject the hypothesis of equal means, and present the results.
8. Also present and discuss the confusion matrices of the classification results.
9. Discuss the limitations of your methodology and threats-to-validity of experimental results.
10. Discuss future work and implications for the research field.
Author Response
Dear Reviewer,
Thank you very much for your valuable comments.
The responses are attached as PDF.

Reviewer 2 Report
I have several questions:
1.- Figures can be improved
2.- The use of the two data sets is not entirely clear. According to the text, in apt. 2.1, it seems that one of them is used for training and the other for validation. Later in section 2.5, it is said that cross validation has been used. Finally, in section 3.2, "validation step" is mentioned.
It should be made clear, which subset of data has been used in each step.
3.- Has any subset of data totally independent of training and validation been used to make a test?
4.- Are the results in Table 1 from training/validation or from testing with a completely independent subset of data?
5.- Has any other method of feature selection been tried?
Author Response

(The authors gave the same response as above.)

Round 2
Reviewer 1 Report
I found the authors' response to my comments unsatisfactory. Only minor edits were done, which is below the level of corrections required by Major Revision. I refer the authors to my previous review, if the authors would like to continue revising this paper in another round of revisions.
Author Response
Dear Reviewer,
We feel that the second version of the manuscript after applying reviewer's comments became more professional and we appreciate the comments. Regarding making it better, we need to know which parts of the answers are not satisfied. It will definitely help us narrow our target for revision.
Best Regards,
